# Sb$_2$S$_3$-templated synthesis of sulfur-doped Sb-N-C with hierarchical architecture and high metal loading for H$_2$O$_2$ electrosynthesis

Minmin Yan[1,5], Zengxi Wei[2,5], Zhichao Gong[1], Bernt Johannessen [3], Gonglan Ye [1]✉, Guanchao He[1], Jingjing Liu[1], Shuangliang Zhao [2]✉, Chunyu Cui [1]✉ & Huilong Fei [1,4]✉

Selective two-electron (2e⁻) oxygen reduction reaction (ORR) offers great opportunities for hydrogen peroxide (H$_2$O$_2$) electrosynthesis and its widespread employment depends on identifying cost-effective catalysts with high activity and selectivity. Main-group metal and nitrogen coordinated carbons (M-N-Cs) are promising but remain largely underexplored due to the low metal-atom density and the lack of understanding in the structure-property correlation. Here, we report using a nanoarchitectured Sb$_2$S$_3$ template to synthesize high-density (10.32 wt%) antimony (Sb) single atoms on nitrogen- and sulfur-codoped carbon nanofibers (Sb-NSCF), which exhibits both high selectivity (97.2%) and mass activity (114.9 A g⁻¹ at 0.65 V) toward the 2e⁻ ORR in alkaline electrolyte. Further, when evaluated with a practical flow cell, Sb-NSCF shows a high production rate of 7.46 mol g$_{catalyst}$⁻¹ h⁻¹ with negligible loss in activity and selectivity in a 75-h continuous electrolysis. Density functional theory calculations demonstrate that the coordination configuration and the S dopants synergistically contribute to the enhanced 2e⁻ ORR activity and selectivity of the Sb-N$_4$ moieties.

Electrochemical oxygen reduction reaction (ORR) is essential in various processes of clean energy conversion and utilization[1,2]. When proceeding via a 4e⁻ pathway (alkaline condition: $O_2 + 2H_2O + 4e^- \rightarrow 4OH^-$), the ORR reduces $O_2$ into $H_2O$ that is preferred in fuel cells and metal-air batteries for maximized energy conversion efficiency[3,4]. In comparison, for the 2e⁻ ORR pathway (alkaline condition: $O_2 + H_2O + 2e^- \rightarrow HO_2^- + OH^-$), it transforms $O_2$ into hydrogen peroxide (H$_2$O$_2$), which is a valuable chemical widely used in disinfection, wastewater treatment, bleaching, chemical synthesis, and other applications[5–7]. The 2e⁻ ORR process enables the portable, on-demand, and distributed synthesis of H$_2$O$_2$, making it a highly promising replacement for the traditional energy-intensive anthraquinone oxidation/reduction process[8,9]. The

development of high-performance catalysts toward the 2e⁻ ORR is the key to realizing the scalable electrosynthesis of H$_2$O$_2$. While noble metal-based electrocatalysts (for example, Pd-Au, Pt-Hg, Pd-Hg) are highly efficient for this process, the high cost and scarcity of noble metals impede their widespread employment[10–13]. Thus, it is imperative to develop cost-effective alternatives to catalyzing the 2e⁻ ORR with high activity and selectivity.

Heterogeneous single-atom catalysts (SACs) consisting of metal atoms dispersed on a solid support have attracted tremendous interest for their advantageous features such as high metal utilization efficiency, uniform active sites, and tailorable coordination environments[14–18]. In particular, metal-nitrogen-carbon (M-N-C)

[1]State Key Laboratory for Chemo/Biosensing and Chemometrics, and College of Chemistry and Chemical Engineering, Hunan University, Changsha 410082, China. [2]Guangxi Key Laboratory of Petrochemical Resource Processing and Process Intensification Technology and School of Chemistry and Chemical Engineering, Guangxi University, Nanning 530004, China. [3]Australian Synchrotron, ANSTO, Clayton, VIC 3168, Australia. [4]Advanced Catalytic Engineering Research Center of the Ministry of Education, Hunan University, Changsha 410082, China. [5]These authors contributed equally: Minmin Yan, Zengxi Wei. ✉e-mail: glye@hnu.edu.cn; szhao@gxu.edu.cn; cycui@hnu.edu.cn; hlfei@hnu.edu.cn

materials represent a unique class of SACs that have been widely utilized to catalyze a variety of electrochemical energy conversion processes[19–23]. Among the different metals in M-N-Cs, Fe, Co and Mn are hitherto the most intensively studied since the associated metal-$N_x$ moieties are promising alternatives to platinum-group metals for catalyzing the 4e⁻ ORR[24–27]. Nevertheless, recent studies suggested that the selectivity of M-N-Cs can be effectively steered toward the 2e⁻ ORR by manipulating the metal-support interaction that is influenced by the microenvironments in the first and outer coordination spheres of metal sites. For example, Jung et al. modified the Co-$N_4$ moiety in a Co-N-C catalyst with electron-rich oxygen species to achieve optimal adsorption of the key ORR intermediate *OOH for efficient $H_2O_2$ production[28]. Recently, we found that the $H_2O_2$ selectivity of the Co-N-C catalyst can be significantly enhanced via the synergy of lowered coordination number and the surrounding oxygenated groups[20].

While the investigations on M-N-Cs have focused on *d*-block transition metals, recent studies have demonstrated that main-group (*s*- and *p*-block) metals (for example, Mg, Ca, In, Sn, Sb, Bi), which were typically considered to be catalytically inert due to the closed *d*-band shells, can be activated via engineering the atomic M-$N_x$ moieties, resulting in promising catalytic reactivity[29–37]. Especially for the 4e⁻ ORR process, main-group M-N-Cs have exhibited rivaling activity compared to the state-of-the-art Fe-based counterparts, while offering additional benefits in terms of improved durability by alleviating the transition metal-induced Fenton reactions[38–43]. However, the use of main-group M-N-Cs to catalyze the 2e⁻ ORR has received much less attention due to the lack of efficient strategies in tailoring the microenvironments of the metal sites[31]. On the other hand, the existing main-group M-N-Cs suffer from low metal-atom densities (typically <3 wt%) (Supplementary Table 1), which limits their catalytic performance. Further increasing the metal contents is challenging as the excessive addition of metal precursors could lead to the undesirable aggregation of metal species into clusters or nanoparticles in conventional pyrolysis synthesis. To this end, some strategies have been developed to alleviate the aggregation of metal atoms during pyrolysis, such as the metal molecular grafting, spatial confinement, multilayer stabilization,

and cascade anchoring[44–46]. However, these strategies often require complex steps and their application to main-group metals are yet to be demonstrated. Therefore, the exploration of strategies in regulating the microenvironments and increasing the metal-atom loading is the key to achieving high-performance main-group M-N-Cs toward the 2e⁻ ORR for $H_2O_2$ electrosynthesis.

Here, we develop a main-group M-N-C catalyst for highly efficient 2e⁻ ORR consisting of high-density Sb atoms (~10.32 wt%) supported on hierarchically porous N/S-doped carbon nanofibers (denoted as Sb-NSCF). Sb-NSCF is synthesized via a deliberately designed pyrolysis strategy with nanoarchitectured $Sb_2S_3$ as templates (Fig. 1a), with the following advantageous features: (i) the $Sb_2S_3$ templates provide abundant Sb and S doping sources during pyrolysis, facilitating the achievements in high loadings of Sb metal atoms and S heteroatoms; (ii) the excessive amounts of $Sb_2S_3$, with low sublimation temperature in vacuum, can be easily removed to enable the exclusive atomic dispersion of Sb without post treatment (for example, acid leaching) that is typically required in conventional pyrolysis synthesis; (iii) the removal of the $Sb_2S_3$ templates results in the formation of hollow carbon nanofibers hierarchically bundled into a flower-shaped assembly that would be beneficial for mass transfer and active-site exposure during catalysis. Rotating ring-disk electrode (RRDE) measurements demonstrate that Sb-NSCF exhibits a high $H_2O_2$ selectivity in a wide potential range and a high mass activity. Further, when assembled into a practical flow-cell device, Sb-NSCF displays a large $H_2O_2$ production rate in a long-term continuous electrolysis with negligible degradation. Theoretical calculations reveal that the unique local atomic environments modified with the S dopants can effectively tune the electronic structure of metal sites, enabling the optimal adsorption of the key reaction intermediate *OOH for the electrosynthesis of $H_2O_2$.

## Results
### Synthesis and characterizations of Sb-NSCF
A schematic illustrating the preparation process of Sb-NSCF is depicted in Fig. 1a. Briefly, antimony acetate, thioacetamide, and ethylene

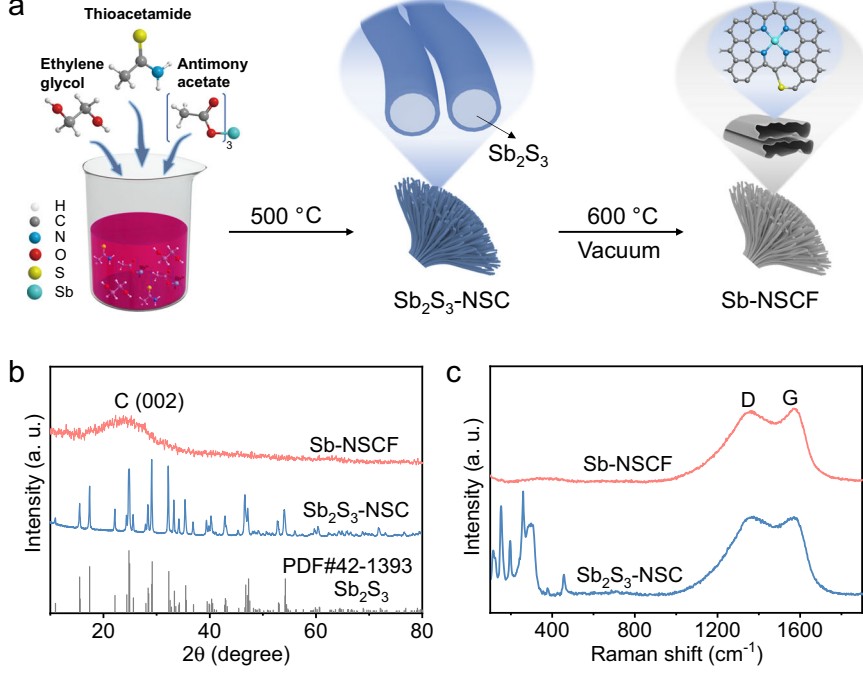

**Fig. 1 | Synthesis and characterizations of Sb-NSCF. a** Schematic illustration of the preparation of Sb-NSCF. **b** XRD patterns of Sb-NSCF, $Sb_2S_3$-NSC, and the $Sb_2S_3$ standard. **c** Raman spectra of Sb-NSCF and $Sb_2S_3$-NSC.

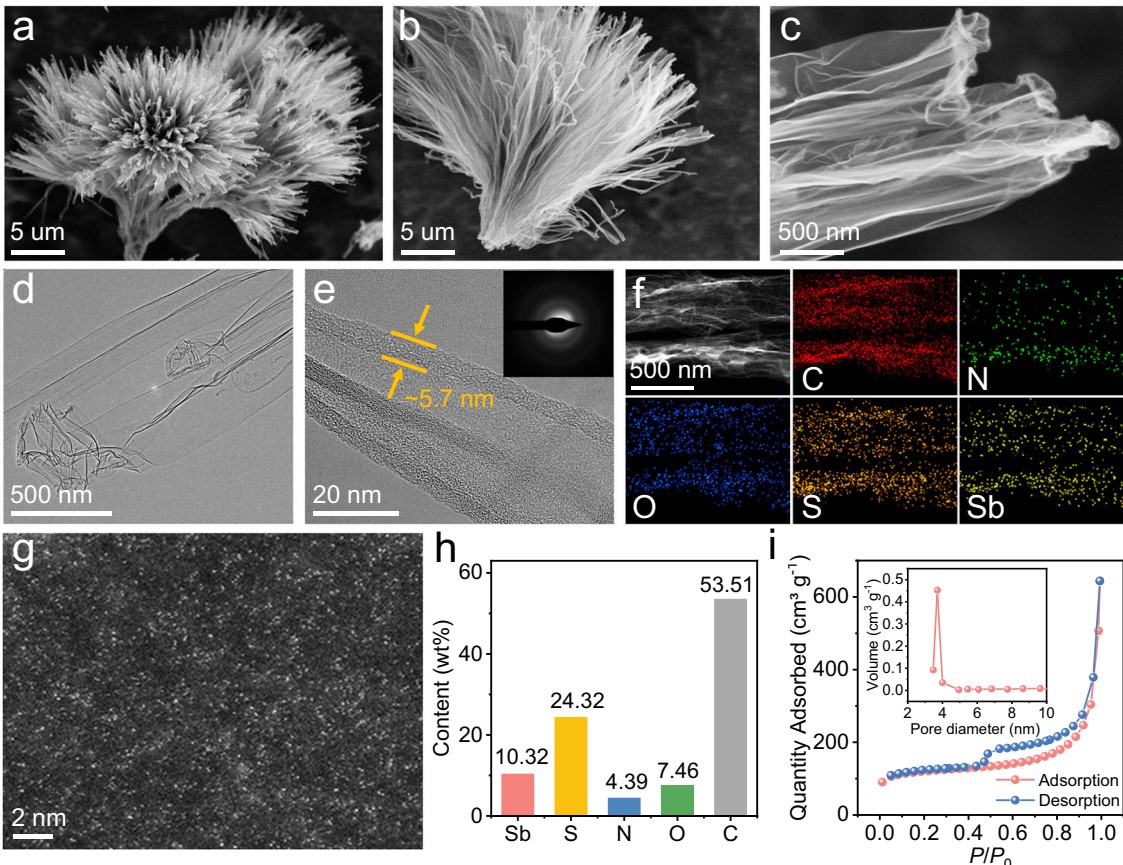

**Fig. 2 | Characterizations of the morphology and structure of Sb-NSCF. a** Top-view SEM image of Sb-NSCF. **b** Side-view SEM image of Sb-NSCF. **c** Enlarged view in (**b**), showing the morphology of the hollow carbon nanofibers. **d**, **e** TEM images of Sb-NSCF at different magnifications. Inset of **e** shows the SAED of Sb-NSCF, suggesting the amorphous nature of the carbon shells. The thickness of the carbon shells is ~5.7 nm. **f** EDS elemental mapping of Sb-NSCF, suggesting the uniform distributions of C, N, O, S, and Sb elements. **g** HAADF-STEM images of Sb-NSCF, revealing the dense dispersion of the Sb single atoms. **h** Chart showing the percentages of Sb, S, N, O, and C elements in Sb-NSCF measured by XPS. **i** N$_2$ adsorption-desorption isotherms of Sb-NSCF with the inset showing the corresponding pore size distribution. Sb-NSCF has a BET-specific surface area of 445.4 m$^2$ g$^{-1}$.

glycol were mixed and vigorously stirred to form a reddish suspension, which was then transferred into a tube furnace and calcinated at 500 °C in an inert atmosphere to obtain a core-shelled structure with Sb$_2$S$_3$ nanorods encapsulated in carbon sheaths (denoted as Sb$_2$S$_3$-NSC). After that, Sb$_2$S$_3$-NSC was further calcinated at 600 °C under vacuum to remove the Sb$_2$S$_3$ nanorods, resulting in the formation of Sb-NSCF. It is noted that the calcination temperature had been optimized by evaluating the catalytic performances of samples calcinated at different temperatures (Supplementary Fig. 1 and Supplementary Fig. 2). The relatively low calcination temperature of 600 °C is beneficial for the lower energy intensive synthesis and the achievement of high metal loadings by mitigating the high-temperature aggregation of metal atoms. The detailed description for the synthesis of Sb-NSCF was provided in the Experimental Section. As revealed by the X-ray diffraction (XRD) patterns, Raman spectra, and transmission electron microscopy (TEM) images of Sb$_2$S$_3$-NSC (Fig. 1b, c and Supplementary Fig. 3), the Sb$_2$S$_3$ nanorods in the core are crystalline and pure in phase. In strong comparison, the XRD patterns and Raman spectra of Sb-NSCF display only spectral signals attributed to carbon without any noticeable contribution from Sb$_2$S$_3$ (Fig. 1b, c), confirming their complete removal after the second calcination step. Further, the broad XRD peak at ~23.6° assigned to the C (002) plane and the large Raman D-to-G band ratio ($I_D/I_G$ = 0.95) of Sb-NSCF imply that the carbon sheaths are highly defective[47].

The top-view scanning electron microscopy (SEM) image in Fig. 2a reveals that Sb-NSCF possesses a flower-like architecture, inherited

from the Sb$_2$S$_3$-NSC template (Supplementary Fig. 4). Side-view SEM image displays that the flower-like structure is comprised of a cluster of nanofibers (10–20 μm in length) bunched at one end and splayed out at the other (Fig. 2b and Supplementary Fig. 5). The nanofibers are hollow and thin in wall thickness and due to the high aspect ratio and flexibility they tend to collapse together, as revealed by the enlarged SEM image (Fig. 2c). The TEM image in Fig. 2d confirms this observation and further demonstrates that the surface of Sb-NSCF is clean with no presence of crystalline metal nanoparticles, consistent with the XRD and Raman results. The carbon nanofiber has a wall thickness of ~5.7 nm (Fig. 2e), and is in amorphous nature as suggested by the diffused halo ring of the selected-area electron diffraction (SAED) (inset of Fig. 2e). The amorphous feature of the carbon substrate in Sb-NSCF can be ascribed to the fact that different from transition metals, the main-group Sb metal is inactive for catalyzing the graphitization of carbon during the pyrolysis synthesis and it could lead to unique catalytic reactivity via the metal-support interaction. The energy-dispersive X-ray spectroscopy (EDS) elemental mapping of Sb-NSCF reveals the uniform distribution of Sb, S, N, O, and C elements throughout the entire carbon nanofiber (Fig. 2f). High-angle annular dark-field scanning TEM (HAADF-STEM) images suggest that the Sb single atoms, represented by the bright dots, are densely dispersed on the carbon matrix (Fig. 2g and Supplementary Fig. 6). X-ray photoelectron spectroscopy (XPS) analysis indicates that Sb-NSCF is composed of Sb (10.32 wt%), S (24.32 wt%), N (4.39 wt%), O (7.46 wt%), and C (53.51 wt%), as summarized in Fig. 2h. The Sb content in Sb-NSCF was

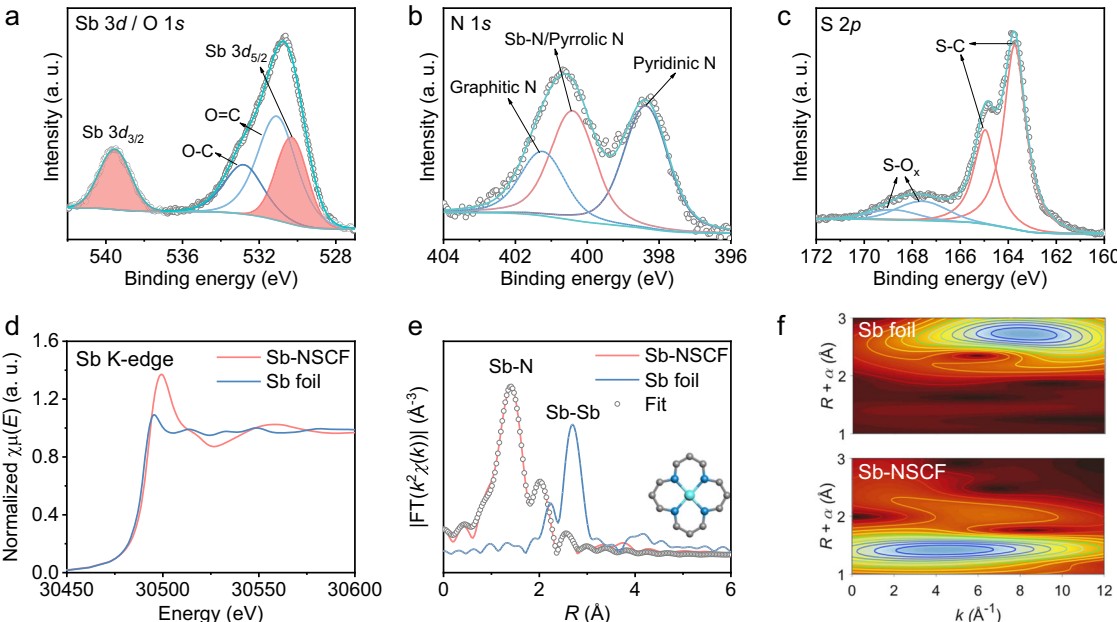

**Fig. 3 | Atomic and electronic structure of Sb-NSCF. a–c** High-resolution XPS spectrum of Sb 3*d*/O 1*s*, N 1*s*, and S 2*p*, respectively. **d** The Sb K-edge XANES spectra of Sb-NSCF and Sb foil. **e** The *k*²-weighted FT-EXAFS spectra of different samples and FT-EXAFS fitting plots of Sb-NSCF. The inset is the structural model of Sb-NSCF. **f** WT-EXAFS plots of Sb-NSCF and Sb foil.

further measured by the thermogravimetric analysis (TGA) to be ~10.1 wt% (Supplementary Fig. 7), which was significantly higher than those of previously reported main-group SACs (Supplementary Table 1). The nitrogen adsorption-desorption isotherms determine that Sb-NSCF has a Brunauer-Emmett-Teller (BET) specific surface area of 445.4 m² g⁻¹ and the pore size distributions suggest the formation of mesopores with the pore size centered at ~3.7 nm (Fig. 2i). The abundance of mesopores could contribute to the enhanced exposure of active sites and mass transport during catalysis. In addition, Sb-NSCF possesses a large electrochemically active surface area (ESCA) of 59.6 m² g⁻¹, estimated from the double-layer capacitance ($C_{dl}$) in the non-Faradaic potential region (Supplementary Fig. 8).

## Analysis of atomic and electronic structure

The atomic and electronic structure of Sb species were analyzed by XPS, X-ray absorption near-edge structure (XANES), and extended X-ray absorption fine structure (EXAFS). As shown in the high-resolution Sb 3*d*/O 1*s* XPS spectrum of Sb-NSCF (Fig. 3a), the peaks assigned to the Sb $3d_{5/2}$ and Sb $3d_{3/2}$ are located at the binding energy of around 530.3 eV and 539.5 eV, respectively, which are higher than those (528.6 eV and 538.0 eV) of Sb metals, indicating that the Sb is in ionic state ($Sb^{\delta+}$)[33,48]. The N 1*s* XPS spectrum in Fig. 3b can be deconvoluted into pyridinic N (398.4 eV), Sb-N/pyrrolic N (400.4 eV), and graphitic N (401.2 eV)[49,50]. The S 2*p* XPS spectrum can be fitted into oxidized S (167.6 eV and 168.8 eV) and S-C (163.7 eV and 165.0 eV) (Fig. 3c)[51], indicating the successful doping of S heteroatoms into the carbon lattices. It is noted that the XPS peak for the S-Sb bonding observed in Sb₂S₃-NSC is absent in Sb-NSCF (Supplementary Fig. 9), suggesting the complete evaporation of Sb₂S₃ species during the second-step pyrolysis. The deconvolution of the C 1*s* XPS spectrum was provided in Supplementary Fig. 10. Figure 3d displays the Sb K-edge XANES profiles of Sb-NSCF and Sb foil as the reference sample. The results show that the XANES profiles of Sb-NSCF and Sb foil are significantly different in shape and that the Sb K-edge position of Sb-NSCF is located at a higher energy compared to that of Sb foil, suggesting that the Sb atoms in Sb-NSCF are positively charged[48], consistent with the XPS analysis. In the Fourier transform EXAFS (FT-EXAFS) (Fig. 3e), Sb-NSCF exhibits a prominent peak at ~1.44 Å that can be assigned to the atomic Sb coordinated to light heteroatoms (for example, N), while no Sb-Sb bonding at 2.69 Å was detected, indicating the Sb species in Sb-NSCF are atomically dispersed. Wavelet transform EXAFS (WT-EXAFS) of Sb-NSCF displays a strong signal contour plot focused at 4.3 Å⁻¹, distinctly different from Sb foil (8.1 Å⁻¹), confirming the atomic dispersion of Sb in Sb-NSCF (Fig. 3f). The quantitative least-squares *R*-space and *k*-space EXAFS curve-fitting analysis was carried out to investigate the coordination configuration of Sb in Sb-NSCF (Fig. 3e, Supplementary Fig. 11 and Supplementary Table 2). The best fitting results for the first coordination shell determine that the Sb atom is coordinated with 3.9 N atoms on average, suggesting the adoption of the Sb-N₄ configuration (inset in Fig. 3e).

## Electrocatalytic H₂O₂ production

The ORR electrocatalytic performances of Sb-NSCF were first evaluated in O₂-saturated 0.1 M KOH solution using the typical RRDE technique (Supplementary Fig. 12) with an optimized catalyst loading of ~20.2 µg cm⁻² (Supplementary Fig. 13). The collection efficiency of RRDE was calibrated to be 0.365 (Supplementary Fig. 14). All potentials reported in this work were converted to a reversible hydrogen electrode (RHE) according to the calibration result (Supplementary Fig. 15). To study the roles of the Sb and S atoms in affecting the catalytic reactivity of Sb-NSCF, control samples of metal-free NSC and Sb-NCF (with a low S content) were prepared. NSC was prepared following the same procedure with Sb-NSCF except that no Sb precursor (antimony acetate) was added, while Sb-NCF was prepared by increasing the second-step calcination temperature to 1000 °C. The higher calcination temperature resulted in significant decrease (from 24.32 wt% to 10.86 wt% by XPS) of S dopants in Sb-NCF (Supplementary Fig. 16). The detailed characterizations regarding the composition and structure of NSC and Sb-NCF were provided in Supplementary Fig. 17–20. Figure 4a shows the linear sweep voltammetry (LSV) curves for the ORR process collected at 1600 rpm, together with the H₂O₂ oxidation current density collected by the Pt ring electrode poised at a constant potential of 1.20 V versus RHE. It is apparent that Sb-NSCF exhibits a superior ORR activity to Sb-NCF and NSC, indicated by its significantly larger current densities at both the

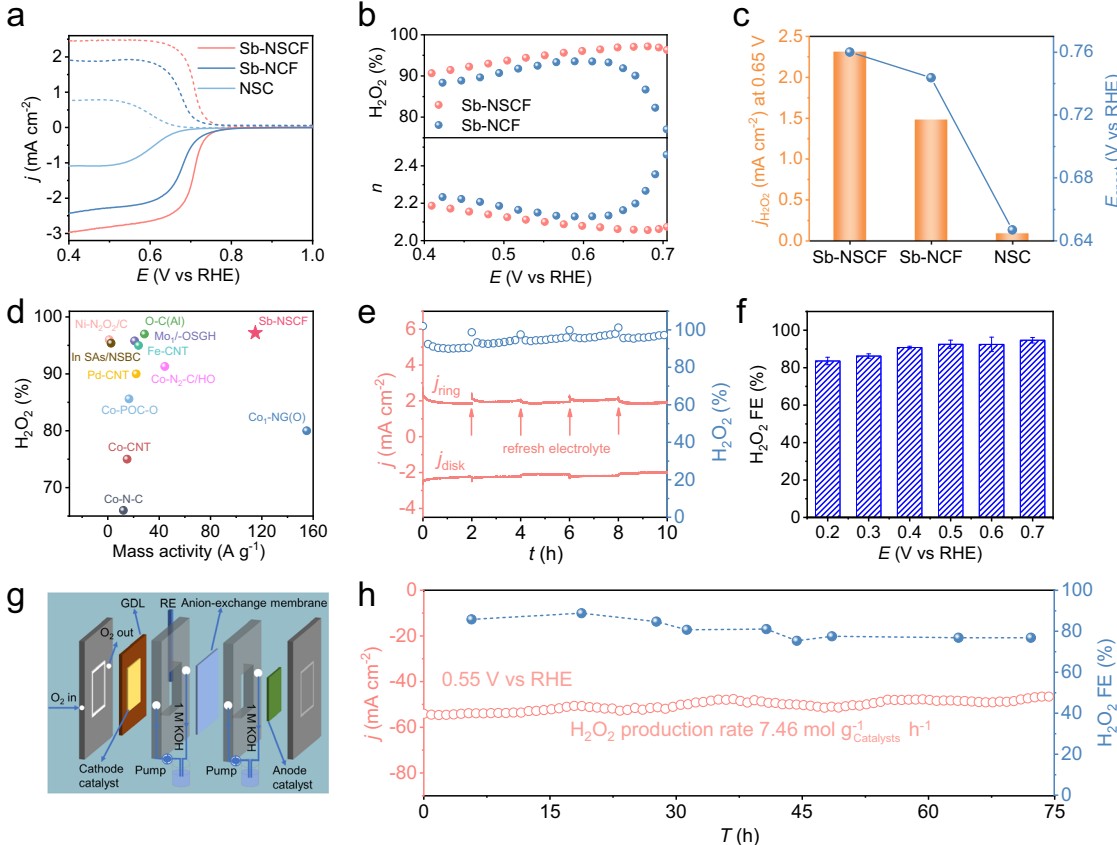

**Fig. 4 | Electrocatalytic ORR performance. a** Electrochemical oxygen reduction polarization curves (solid lines) at a rotation of 1600 rpm and simultaneous $H_2O_2$ detection currents on the ring electrode (dashed lines) for Sb-NSCF, Sb-NCF, and NSC in $O_2$-saturated 0.1 M KOH electrolyte. **b** Calculated $H_2O_2$ selectivity (%) and electron transfer number ($n$) on Sb-NSCF and Sb-NCF based on the RRDE measurements. **c** Comparison of $H_2O_2$ current density ($j_{H_2O_2}$) at 0.65 V and onset potential at $j_{H_2O_2}$ = 0.1 mA cm$^{-2}$ for Sb-NSCF, Sb-NCF, and NSC. **d** Comparison of the maximum $H_2O_2$ selectivity and mass activity at 0.65 V for $H_2O_2$ production in alkaline media between Sb-NSCF and previously reported SACs in Supplementary

Table 3. **e** Chronoamperometry stability test of Sb-NSCF at 0.65 V in 0.1 M KOH. The Pt ring electrode was cleaned by cyclic voltammetry scanning from 0 V to −0.3 V versus RHE for 40 cycles and the electrolyte was refreshed every 2 h during the test. **f** The $H_2O_2$ Faradaic efficiency (%) of Sb-NSCF at different potentials determined by the ceric sulfate titration method using the H-cell setup. **g** Illustration of the three-phase flow cell setup for $H_2O_2$ production. **h** Chronoamperometry stability test of Sb-NSCF at 0.55 V for the $H_2O_2$ production in flow cell using 1 M KOH as electrolyte and the corresponding $H_2O_2$ Faradaic efficiency.

disk and ring electrodes. The calculated $H_2O_2$ selectivity (%) and electron transfer number ($n$) as a function of potential were plotted in Fig. 4b. In a wide potential range from 0.40 V to 0.70 V, the $H_2O_2$ selectivity of Sb-NSCF catalyst is larger than 90% (maximized at 97.2%) and the $n$ is lower than 2.2, highlighting a highly selective 2e$^-$ ORR pathway. In comparison, Sb-NCF shows much lower $H_2O_2$ selectivity and higher $n$ values, demonstrating the importance of S dopants in tuning the ORR selectivity. As reference samples, the commercial Pt/C catalyst exhibited a diffusion-limited disk current density close to 6.0 mA cm$^{-2}$ with negligible ring current density and $H_2O_2$ production (Supplementary Fig. 21), suggesting that it catalyzed the ORR via the 4e$^-$ process. $H_2O_2$ reduction reaction measurements suggested that Sb-NSCF exhibited an insignificant $H_2O_2$ reduction current compared to Pt/C (Supplementary Fig. 22), which could avoid the further reduction of the $H_2O_2$ product[20]. Besides, Sb-NSCF delivers the most positive onset potential ($E_{onset}$) of 0.76 V (defined as the potential at the ring current density of 0.1 mA cm$^{-2}$) and a highest $H_2O_2$ current density ($j_{H_2O_2}$) of 2.31 mA cm$^{-2}$ at 0.65 V (Fig. 4c), superior to those of Sb-NCF (0.74 V, 1.48 mA cm$^{-2}$) and NSC (0.65 V, 0.09 mA cm$^{-2}$). The kinetic current density ($j_k$) calculated by the Koutecký–Levich equation shows that Sb-NSCF possesses a highest value of 32.6 mA cm$^{-2}$ at 0.65 V, vastly exceeding that of Sb-NCF (4.9 mA cm$^{-2}$) and NSC (0.1 mA cm$^{-2}$), manifesting its faster ORR kinetics for $H_2O_2$ production (Supplementary Fig. 23). Furthermore, the analysis of Tafel plots reveals that Sb-NSCF

possesses a Tafel slope of 29.4 mV dec$^{-1}$ (Supplementary Fig. 24), which was much smaller than those of Sb-NCF (42.4 mV dec$^{-1}$) and NSC (60.5 mV dec$^{-1}$), indicating its more favorable ORR kinetics. The electrochemical impedance spectroscopy (EIS) plots present a smaller charge transfer resistance for Sb-NSCF than the control catalysts (Supplementary Fig. 25), confirming its fast reaction kinetics that could be attributed to the high density of Sb atoms and unique microenvironments, as discussed later. The important role of the Sb sites was emphasized by the poison study (Supplementary Fig. 26), showing that the ORR activity of Sb-NSCF declined obviously upon the addition of thiocyanate ions (SCN$^-$). Remarkably, this main-group Sb-NSCF catalyst with high $H_2O_2$ selectivity (97.2%) and mass activity (114.9 A g$^{-1}$ at 0.65 V) is superior to almost all of the previously reported SACs (Fig. 4d and Supplementary Table 3). Moreover, Sb-NSCF shows stable activity and selectivity, as demonstrated by the 10-h chronoamperometry test and 10,000 scans of cyclic voltammetry (CV) test (Fig. 4e and Supplementary Fig. 27).

To explore the possibility for practical use, Sb-NSCF was casted on a hydrophobic carbon fiber paper and assembled in an H-cell electrolyzer (Supplementary Fig. 28) containing $O_2$-saturated 0.1 M KOH solution. To determine the Faradaic efficiency of Sb-NSCF, the working electrode was biased at a constant potential for 10 min and the accumulated amounts of $H_2O_2$ in the electrolyte of the cathode compartment were determined by a colorimetric quantification method.

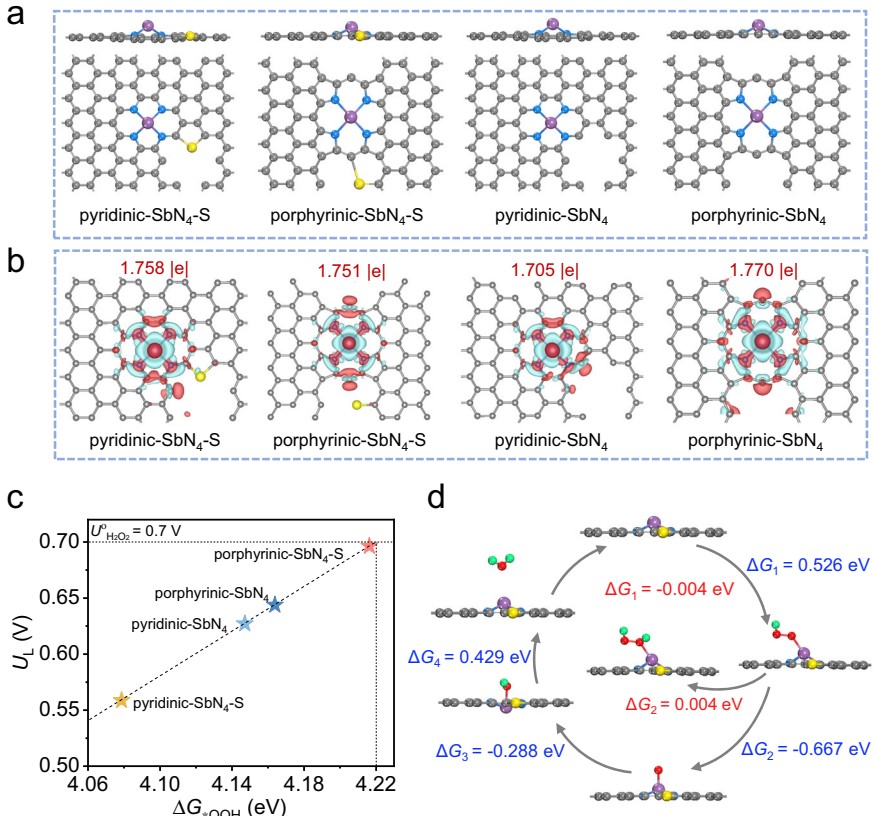

**Fig. 5 | Theoretical analysis. a** The side and top views of optimized structural models of pyridinic-SbN$_4$-S, porphyrinic-SbN$_4$-S, pyridinic-SbN$_4$, and porphyrinic-SbN$_4$. The gray, blue, yellow, and purple spheres represent C, N, S, and Sb atoms, respectively. **b** The calculated charge density difference for the corresponding structural models in (**a**). The red and cyan isosurfaces (0.003 Bohr$^{-3}$) represent the electron gain and loss, respectively. **c** Limiting potentials ($U_L$) as a function of $\Delta G_{*OOH}$ for the four models. **d** The 2e$^-$ and 4e$^-$ ORR reaction pathways on the porphyrinic-SbN$_4$-S configuration. The green, grey, blue, red, yellow, and purple spheres represent H, C, N, O, S, and Sb atoms, respectively.

The results in Fig. 4f show that the Faradaic efficiency of Sb-NSCF for H$_2$O$_2$ generation is between 83.6% and 94.7% in the wide potential range of 0.20–0.70 V. Further, a three-phase flow cell setup with a gas diffusion electrode (GDE) (Fig. 4g and Supplementary Fig. 29) was utilized to enhance the H$_2$O$_2$ productivity by circumventing the issue of the low solubility of O$_2$ in aqueous electrolyte. The LSV obtained from the GDE shows much higher current densities as compared from those measured by RRDE (Supplementary Fig. 30). Significantly, during the bulk electrolysis at 0.55 V (Fig. 4h), a high current density of ~50 mA cm$^{-2}$ was maintained without noticeable degradation for at least 75 h, enabling a continuous and stable production of H$_2$O$_2$, which is important for industrial practices. Moreover, the GDE exhibited an averaged H$_2$O$_2$ Faradaic efficiency of ~80% and a high production rate of 7.46 mol g$_{catalyst}^{-1}$ h$^{-1}$, which are amongst the highest values reported so far (Supplementary Table 3). These results suggest that Sb-NSCF is a promising candidate for H$_2$O$_2$ electrosynthesis at the industrial level.

## Theoretical calculations

Density functional theory (DFT) calculations were performed to gain insights into the origin of the enhanced catalytic reactivity and the reaction mechanism of Sb-NSCF. Given that the coordinated nitrogen types of the metal centers and the modification of the carbon substrate can both affect the electronic structure and thus catalytic reactivity of the active metal sites[52,53], we considered four structural models of Sb-N-C catalysts adopting the pyridinic and porphyrinic configurations with and without the S dopants, namely pyridinic-SbN$_4$-S, porphyrinic-SbN$_4$-S, pyridinic-SbN$_4$, and porphyrinic-SbN$_4$, as depicted in Fig. 5a. The optimized structures for these four models reveal that the Sb atoms are

located out of the carbon plane and that the introduce of S dopants barely changes the geometries of the Sb-N$_4$ configurations. Figure 5b shows the differences in the charge state of Sb atoms in the four models. The Bader charge of Sb atom on the pyridinic-SbN$_4$ and porphyrinic-SbN$_4$ are 1.705|e| and 1.770|e|, respectively. After introducing the S dopants, the value changes to 1.758|e| and 1.751|e| for the pyridinic-SbN$_4$-S and porphyrinic-SbN$_4$-S, respectively. These results indicate that the types of the coordination nitrogen and the outer-sphere S atoms could modulate the charge state of Sb, thereby influencing its interaction with the ORR intermediate *OOH. Since the formation free energy of the intermediate *OOH ($\Delta G_{*OOH}$) is typically considered as a good descriptor to determine the ORR reactivity of a catalyst[54], we calculated the thermodynamic limiting potential ($U_L$) for the H$_2$O$_2$ formation as a function of $\Delta G_{*OOH}$ on the four models (Fig. 5c). $U_L$ is defined as the maximum potential at which all the reaction steps are downhill in free energy and therefore can be considered as a metric of activity. Among the four models studied, porphyrinic-SbN$_4$-S has the largest $\Delta G_{*OOH}$ of 4.216 eV, suggesting the weak adsorption strength of the *OOH intermediate on the Sb site and thus its facile removal to form the product of H$_2$O$_2$. Correspondingly, porphyrinic-SbN$_4$-S exhibits the highest $U_L$ of 0.696 V, manifesting its highest catalytic activity toward the 2e$^-$ ORR. To gain further insights into the catalytic selectivity, the energy barriers associated with each step for both the 2e$^-$ and 4e$^-$ reaction pathways on the porphyrinic-SbN$_4$-S were calculated, as shown in Fig. 5d. For the 4e$^-$ pathway, the rate-determining step (RDS) is O$_2$(g) + * + H$^+$ + e$^-$ $\xrightarrow{\Delta G_1}$ *OOH, with an energy barrier of 0.562 eV. In comparison, the RDS (*OOH + H$^+$ + e$^-$ $\xrightarrow{\Delta G_2}$ H$_2$O$_2$) for the 2e$^-$ pathway has a much smaller energy barrier of 0.004 eV, indicating that the 2e$^-$

pathway is more energetically favorable on the porphyrinic-SbN$_4$-S. These calculation results suggest that the coordination configuration and the introduce of S dopants could synergistically contribute to the enhanced activity and selectivity of the Sb-N$_4$ moieties towards the 2e$^-$ ORR, corroborating the experimental observations.

## Discussion

In summary, we have developed a unique Sb$_2$S$_3$-templated strategy to fabricate high-density main-group Sb atoms dispersed in N,S-codoped hollow carbon nanofibers that are further assembled into a flower-like architecture. Sb-NSCF exhibited high ORR performances for H$_2$O$_2$ production with an early onset potential (0.76 V at 0.1 mA cm$^{-2}$), high H$_2$O$_2$ selectivity (up to 97.2%), and mass activity (114.9 A g$^{-1}$ at 0.65 V). In addition, the H$_2$O$_2$ production rate was estimated to be 7.46 mol g$_{catalyst}^{-1}$ h$^{-1}$ in a GDE flow-cell test and the electrode maintained good stability without significant decrease in the activity and H$_2$O$_2$ Faradaic efficiency during the 75-h continuous electrolysis, outperforming most of the state-of-the-art catalysts. Combined experimental and theoretical studies suggest that the superior catalytic performance of Sb-NSCF can be attributed to the dense Sb atoms, hierarchical porosity, and the S-doping regulated microenvironments of the Sb-N$_4$ moieties. This study opens an attractive avenue to design highly efficient main-group SACs for the 2e$^-$ ORR and other electrocatalytic processes.

## Methods

### Materials

Antimony acetate, thioacetamide, and ethylene glycol were purchased from Shanghai Macklin Biochemical Co., Ltd, while other chemicals were purchased from Sigma-Aldrich. All the chemicals were directly used without further processing.

### Preparation of Sb-NSCF

First, 4.0 g of antimony acetate and 4.0 g of thioacetamide were dissolved in 4 mL (4.4 g) of ethylene glycol and the mixture was stirred for 24 h to form a reddish homogeneous suspension. Thereafter, the suspension was loaded into a porcelain boat and placed in the central zone of a tube furnace. After purging the tube with N$_2$ flow, the temperature of the tube furnace was increased to 500 °C with a ramping rate of 3 °C min$^{-1}$ and the sample was annealed at 500 °C for 5 h under N$_2$. After being naturally cooled down to room temperature, Sb$_2$S$_3$-NSC was obtained. Sb$_2$S$_3$-NSC was transformed into Sb-NSCF after being further annealed at 600 °C (with a ramping rate of 3 °C min$^{-1}$) for 1 h in a tube furnace under the vacuum condition (~67.5 Pa). The yield of Sb-NSCF was ~0.0347 g. The weight loss can be mainly ascribed to the evaporation of the molecular precursors with low boiling points during the calcination process.

### Preparation of Sb-NCF

Sb-NCF was prepared with the same process as that of Sb-NSCF except that the annealing temperature in the second-step calcination was changed to 1000 °C.

### Preparation of NSC

NSC was prepared with the same process as that of Sb-NSCF except that no antimony acetate was added in the precursor solution.

### Material characterizations

SEM was conducted on a COXEM EM-30 operated at 2 kV and Plus Sigma ZEISS operated at 2 kV. TEM was carried out on a JEOL JEM-2100F microscope operated at an accelerating voltage of 200 kV. XRD analyses were carried out on a Bruker D8 Advance diffractometer with Cu Kα radiation ($\lambda \approx 1.54$ Å). Raman spectra were recorded on a Thermo Scientific DXR Raman microscope with a 532 nm laser. XPS were collected using a Thermo Scientific ESCALAB 250Xi system with

Al Kα radiation ($hv = 1486.6$ eV). TG was conducted on a Netzsch STA 2500 with Al$_2$O$_3$ pan. Nitrogen adsorption-desorption measurements were conducted on a Micromeritics ASAP 2460 Plus system at 77 K. HAADF-STEM and EDX mapping were obtained on a spherical aberration-corrected transmission electron microscope (FEI Titan Cubed Themis G2 300) which was operated at 200 kV. Sb K-edge XAS measurements were performed at beamline Hutch B-standard experimental station at the Australian Synchrotron. The data analysis for the X-ray absorption spectroscopy using IFEFFIT was conducted using the Demeter system.

### Electrochemical measurements

The ORR performances were evaluated with a three-electrode cell equipped with a RRDE (Pine Research Instrumentation, USA) and CHI 760E electrochemical workstation. A Pt wire was used as the counter electrode and a Hg/HgO reference electrode filled with 1 M KOH was used for the measurements in the 0.1 M KOH electrolyte. To exclude the possible influence of Pt contamination from the Pt counter electrode on the performances, a graphite counter electrode was also used for electrochemical measurements (Supplementary Fig. 31). The potential was converted to the RHE scale by $E_{RHE} = E_{Hg/HgO} + 0.896$ V based on the calibration results (Supplementary Fig. 15). For preparing the working electrode, the catalyst ink was prepared by dispersing 1.0 mg of catalyst into 1.98 mL ethanol and 20 μL Nafion (5 wt%) solution and sonicating for 30 min. After polishing RRDE mechanically with alumina suspension, 10 μL of catalyst ink was drop-casted onto the disk electrode (0.2475 cm$^2$ area) of RRDE, resulting a catalyst loading of 20.2 μg cm$^{-2}$ and then the electrode was dried under an infrared lamp to give a uniform catalyst layer. Prior to the ORR measurement, CV scans at a rate of 50 mV s$^{-1}$ from 1.2 V to 0 V were performed until a stable state was reached. Subsequently, the Pt ring was electrochemically cleaned from 0 V to −0.3 V for 50 cycles at a scan rate of 100 mV s$^{-1}$. Then, the LSV polarization curves were scanned from 1.0 V to 0 V at a rate of 10 mV s$^{-1}$ with the RRDE rotating at 1600 rpm. The Pt ring electrode was held at 1.2 V to quantify the amounts of H$_2$O$_2$ produced on the disk electrode. The LSV curves were corrected with 95% $iR$-compensation and background non-Faradaic currents that were obtained by conducting the measurements in the N$_2$-saturated electrolyte. The ring collection efficiency ($N$) of RRDE was determined to be 0.365 using a typical redox couple of potassium ferricyanide solution (Supplementary Fig. 14). The H$_2$O$_2$ selectivity was calculated using the following equation:

$$\text{H}_2\text{O}_2 \text{ selectivity } (\%) = 200 \times \frac{\frac{I_R}{N}}{|I_D| + \frac{I_R}{N}} \tag{1}$$

where $I_R$ is the ring current, $I_D$ is the disk current and $N$ is the collection efficiency. The number of electrons transferred ($n$) was calculated using the equation:

$$n = 4 \times \frac{|I_D|}{|I_D| + I_R/N} \tag{2}$$

The kinetic current density ($j_k$) was calculated according to the Koutecký–Levich equation:

$$\frac{1}{j} = \frac{1}{j_l} + \frac{1}{j_k} \tag{3}$$

where $j$ and $j_l$ are the measured current density and diffusion-limited current density for H$_2$O$_2$ production, respectively. The $j$ value was obtained by dividing the ring current by the disk electrode area and the collection efficiency ($N$). The $j_l$ value was taken from the highest value in the $j$ plot measured over the entire potential range investigated (0–1.0 V). To analyze the kinetics of the catalysts for H$_2$O$_2$ production,

the Tafel plots were generated according to the equation:

$$E = a + b \times \log(j_k) \quad (4)$$

where $a$, $E$, $j_k$, and $b$ are the constant, applied potential, kinetic current for $H_2O_2$ production, and Tafel slope, respectively.

Chronoamperometry stability test was conducted under a constant disk electrode potential at 0.65 V in $O_2$-saturated 0.1 M KOH electrolyte with an RRDE rotating speed of 1600 rpm. The Pt ring electrode was cleaned by CV scanning from 0 V to −0.3 V versus RHE for 40 cycles and the electrolyte was refreshed every 2 h during the chronoamperometry test.

The $H_2O_2$ Faradaic efficiency was measured in a two compartment H-cell with Nafion membrane as separator. The cathode and anode compartments were filled with the same volume (30 mL) of 0.1 M KOH solution with continuous $O_2$ bubbling. The working electrode was a hydrophobic carbon paper (1 cm², Toray, TGP-H-060) with a catalyst loading of 0.1 mg cm⁻². A Hg/HgO electrode was utilized as the reference electrode and platinum wire as the counter electrode. The $H_2O_2$ concentration was measured by a traditional cerium sulfate ($Ce(SO_4)_2$) titration method based on the mechanism that a yellow solution of $Ce^{4+}$ would be reduced by $H_2O_2$ to colorless $Ce^{3+}$ ($2Ce^{4+} + H_2O_2 \rightarrow 2Ce^{3+} + 2H^+ + O_2$). The concentration of $Ce^{4+}$ before and after the reaction can be measured by ultraviolet-visible spectroscopy. Standard $Ce(SO_4)_2$ solution (0.5 mM) was prepared by dissolving $Ce(SO_4)_2$ salts into 0.5 M sulfuric acid solution. The calibration curves between absorbance and $Ce^{4+}$ concentration were determined by measuring the absorbance at 317 nm of different $Ce(SO_4)_2$ solutions with known concentrations (0.1–0.5 mM) (Supplementary Fig. 32). After electrolysis for a certain time period, 100 μL of the electrolyte in the cathode chamber after neutralization by 0.5 M sulfuric acid solution was added into the standard $Ce(SO_4)_2$ titrant solution. Based on the linear relationship between the signal intensity and $Ce^{4+}$ concentration, the molar amounts of consumed $Ce^{4+}$ after reaction could be obtained. By this approach, the amounts of $H_2O_2$ produced can be calculated as half the molar amounts of the $Ce^{4+}$ consumed. The Faradaic efficiency was calculated by the following equation:

$$FE\,(\%) = 2 \times \frac{CVF}{Q} \quad (5)$$

where $F$ is the Faraday constant (96,485 C mol⁻¹), $C$ is the concentration of $H_2O_2$, $V$ is the volume of electrolyte and $Q$ is the total charge during the ORR.

The electrosynthesis of $H_2O_2$ by the catalysts was further evaluated using a three-phase flow-cell device. The Sb-NSCF catalyst was deposited on the hydrophobic carbon paper (working area of 1 cm²) with the loading of 0.1 mg cm⁻² to obtain the gas diffusion electrode. A nickel mesh was used as the anode and the Hg/HgO electrode as the reference electrode. During the measurements, the 1 M KOH electrolyte was circulated through the electrochemical cell using a peristaltic pump (flow rate of 2.9 mL min⁻¹). Simultaneously, a continuous flow of $O_2$ was introduced into the cell (20 mL min⁻¹). Potentiostatic testing at 0.55 V was carried out continuously for 75 h to evaluate the catalyst stability and selectivity for $H_2O_2$ production and the generated $H_2O_2$ was quantified using the $Ce^{4+}$ titration method.

## DFT calculations

The spin-polarized DFT calculations for $H_2O_2$ production on single-atom catalysts were carried out employing the projector augmented wave (PAW) potentials, as implemented in the Vienna ab initio Simulation Package (VASP)[55,56]. The generalized gradient approximation (GGA)/Perdew-Burke-Ernzerhof (PBE) level were adopted[57]. The surface Brillouin zone was sampled with a (2 × 2 × 1) Monkhorst-Pack

k-point grid for single-atom catalysts. All the atomic positions were allowed to relax until the forces were less than 0.02 eV/Å, and the electron convergence energy was set to 10⁻⁵ eV. An energy cutoff of 500 was applied. The vacuum was set to 15 Å for all the single-atom catalysts. The DFT-D3 scheme is adopted to correct the van der Waals interaction[58].

To analyze the $H_2O_2$ production performance, the computational hydrogen electrode (CHE) model was adopted[59]. With an applied bias ($U$), the reaction-free energies can be described as:

$$\Delta G = \Delta E + \Delta ZPE - T\Delta S - neU \quad (6)$$

where $n$ is the number of electrons involved in the reaction. The two-electron and four-electron ORR can be described in the following:

Two-electron ORR:

$$O_2(g) + * + (H^+ + e^-) \xrightarrow{\triangle G_1} *OOH \quad (7)$$

$$*OOH + (H^+ + e^-) \xrightarrow{\triangle G_2} H_2O_2(l) \quad (8)$$

Four-electron ORR:

$$O_2(g) + * + (H^+ + e^-) \xrightarrow{\triangle G_1} *OOH \quad (9)$$

$$*OOH + (H^+ + e^-) \xrightarrow{\triangle G_2} *O + H_2O(l) \quad (10)$$

$$*O + (H^+ + e^-) \xrightarrow{\triangle G_3} *OH \quad (11)$$

$$*OH + (H^+ + e^-) \xrightarrow{\triangle G_4} H_2O(l) \quad (12)$$

## Data availability

All data needed to evaluate the conclusions in the paper are presented in the paper and/or the Supplementary Information. Source data are provided with this paper.

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

## Acknowledgements

H.F. acknowledges financial support from the National Natural Science Foundation of China (Grant No. 51902099, 92163116) and the Innovative Research Groups of Hunan Province (Grant No. 2020JJ1001). G.Y. acknowledges support from the Hunan Province Natural Science Foundation (Grant No. 2020JJ4204) and the National Natural Science Foundation of China (Grant No. 22209043). S.Z. acknowledges support from the National Natural Science Foundation of China (Grant No. 22178072). Density functional theory calculations are supported by the Guangxi University, Guangxi Postdoctoral Innovative Talents Support Program and Guangxi Science and Technology Base and Talent Special Project (AD21075015, AD21220017). We are grateful for the support of XAS beamlines at the ANSTO Australian Synchrotron, Victoria, Australia.

## Author contributions

H.F. oversaw the research in different phases and provided regular guidance and suggestions throughout the research. M.Y. and C.C. discovered the procedure, performed the syntheses, part of the structural characterization and electrochemical tests. Z.W. and S.Z. conducted the theoretical studies. Z.G., G.H., and J.L. assisted in catalytic measurements. B.J. performed the XAFS measurements. M.Y., Z.W., G.Y., S.Z., C.C., and H.F. analyzed the results and co-wrote the paper. All authors had an opportunity to comment on the manuscript.

## Competing interests

The authors declare no competing interests.
