## [Peer Review File · Nature Communications]

Sb₂S₃-templated synthesis of sulfur-doped Sb-N-C with hierarchical architecture and high metal loading for H₂O₂ electrosynthesisREVIEWER COMMENTS

Reviewer #1 (Remarks to the Author):

This is an interesting and well executed paper on the synthesis of atomically-dispersed lead-based catalysts on carbon supports, for the generation of hydrogen peroxide in a flow cell with high selectivity. The materials are novel, well characterised and perform well. However, there remain some issues with the manuscript. As such I recommend major revisions addressing the following points.

The language should be improved throughout. For example, "prominently" is misused in the abstract. There are many other examples throughout the text.

Hydrogen peroxide should be written in full somewhere in the abstract - not just the chemical formula.

Page 4, Line 92-101: I don't think that specific results such as selectivity and mass activity should be explicitly mentioned in the introduction. It is better to save these values for the results section.

The synthesis temperature is quite low compared to other methods of generating M-N-C electrocatalysts. This could be emphasised as an additional advantage for lower energy intensive synthesis. How do you expect the low temperature to affect the graphitisation of the the carbon, and will it have an effect on the electrochemistry? Have you systematically varied the synthesis temperature?

What is the yield of the synthesis? What batch size can be produced?

It may be useful to the reader if you include the BET surface area in figure 2i, or in the caption.

Page 7, line 137: I don't think the "hollow" nature of the structures is visible in Figure 2b. Furthermore, by definition, I don't think that nanobelts can be hollow, since belts are 2D structures. You should reconsider how to describe these structures.

I don't think "scattered" is the correct way to describe this structure. I would call this a cluster of nanofibers bunched at one end and splayed out at the other. Are these bunches of nanofibers consistent throughout the sample? Are these images representative? Adding a low magnification image would help confirm this (i.e. much lower magnification than currently available in SI).

Page 7, line 142: "no presence of metal particles" - You should be more specific - I think you are referring to crystalline metallic nanoparticles.

Why do you think that the pore size is narrowly clustered at 4.2 nm? What is the mechanism of formation of these mesopores? How do they contribute to the electrochemistry?

Figure 3b: Pyrrolic nitrogen is less thermodynamically stable than graphitic or pyridinic nitrogen, and therefore it is unlikely that this is the most abundant species present in your sample. In addition, if you have 10% Sn in the sample, I'd expect a Sn-N peak to be present too - where is it? It may appear if you reanalyse the deconvolutions and shift the main peaks to appropriate binding energies. Similarly, if you have 24% sulfur in the carbon, a S-N peak should also be assigned in the N1s. Broadly speaking, the assignments could be more likely pyridinic (398.5 eV), Sn-N (~400 eV), graphitic N / hydrogenated pyridinic (~401 eV). I recommend that you check the following paper regarding interpretation of XPS spectra: J. Vac. Sci. Technol. A 38(3) May/June 2020; doi: 10.1116/1.5135923 (especially Fig. 5).

The electrochemical characterisation is essentially meaningless if you don't provide a suitable independent reference sample for comparison. Usually Pt/C is used as a 4 electron reference. As you measured in alkaline electrolyte and are interested in 2 electron ORR, nickel might be more suitable. Even better would be a commercial M-N-C catalyst such as Pajarito Powder.

It is not good practice to use a platinum counter electrode in the measurement of nominally platinum-free M-N-C electrocatalysts. In general, a graphite counter electrode should be used to avoid contamination.

Reviewer #2 (Remarks to the Author):

This manuscript developed a main-group metal-nitrogen-carbon catalyst consisting of high-density single Sb atoms supported on N/S-codoped carbon nanobelts (Sb-NSCB) for the electrosynthesis of H₂O₂ via the 2e⁻ ORR. The catalyst was synthesized by a unique Sb₂S₃-templated strategy with high metal-atom loading and hierarchical porous nanostructures and it exhibited exceptional catalytic activity, selectivity and stability. Detailed experimental and theoretical studies were conducted to elucidate the origin for the improved catalytic reactivity. The presented results represent significant achievements in the field of single atom electrocatalysts in terms of new synthetic method, fundamental understanding of the catalytic mechanism and impressive performances and will be attractive to broad audience in materials science and chemical science. Overall, I recommend its publication in Nature Communications after minor revision. Some specific comments are provided as follows:

1. Considering that transition metal-nitrogen-carbon materials such as Co-N-C already exhibited promising catalytic activity for the ORR, what are the potential advantages of developing main-group M-N-C catalysts?
2. Increasing the metal loading is important for the applications of SACs. The high-density (10.32 wt%) is a highlight of this work. The authors are suggested to summarize and discuss the synthetic methods of SACs with high metal loading in the introduction part: DOI: 10.1007/s12274-020-3244-4; etc.
3. The calcination temperature of Sb-NSCB was set at 600 °C under vacuum. Had the calcination temperature been optimized? What is the influence of temperature on the catalytic behaviors?
4. The activity of catalyst toward the H₂O₂ reduction reaction is essential in H₂O₂ production. How about the activity of Sb-NSCB toward this reaction?
5. The BET surface area of Sb-NCB should be provided and compare to that of Sb-NSCB.
6. The authors are suggested to enhance the discussion on the active sites at atomic scale: DOI: 10.1007/s12274-022-4371-x; etc.
7. Why is the carbon amorphous in the synthesized catalyst?
8. A more detailed description of the method for determining the generated amounts of H₂O₂ should be needed.

Reviewer #1 (Remarks to the Author):

General comment: *This is an interesting and well executed paper on the synthesis of atomically-dispersed lead-based catalysts on carbon supports, for the generation of hydrogen peroxide in a flow cell with high selectivity. The materials are novel, well characterized and perform well. However, there remain some issues with the manuscript. As such I recommend major revisions addressing the following points.*

Response: We sincerely thank the reviewer for carefully reviewing our manuscript and kind praise about this work. We especially appreciate the insightful comments and welcome the opportunity to address them.

Specific comment #1: *The language should be improved throughout. For example, "prominently" is misused in the abstract. There are many other examples throughout the text.*

Response: Thank you for the valuable comment. We have deleted the word “prominent” in the abstract and modified the sentence as follows (Page 1, Line 21 – 25): “Here, we report....., which exhibits both high selectivity (97.2%) and mass activity (114.9 A g⁻¹ at 0.65 V) toward the 2e⁻ ORR in alkaline electrolyte.” In addition, we have carefully checked the manuscript to improve the language. For example, “remarkable” in Page 2, Line 26 and Page 4, Line 99 was changed to “high”, and “exceptional” was deleted in the conclusion section.

Specific comment #2: *Hydrogen peroxide should be written in full somewhere in the abstract - not just the chemical formula.*

Response: As suggested, hydrogen peroxide has been written in full in the abstract.

Specific comment #3: *Page 4, Line 92-101: I don't think that specific results such as selectivity and mass activity should be explicitly mentioned in the introduction. It is better to save these values for the results section.*

Response: Thank you for the valuable suggestion. As suggested, the specific results such as selectivity, mass activity and H₂O₂ production rate have been removed from the introduction. The modified version is as follows (Page 4, Line 97 – 101): “Rotating ring-disk electrode (RRDE) measurements demonstrate that Sb-NSCF exhibits a high H₂O₂ selectivity in a wide potential range and a high mass activity. Further, when assembled into a practical flow-cell device, Sb-NSCF displays a large H₂O₂ production rate in a long-term continuous electrolysis with negligible degradation.”

Specific comment #4: *The synthesis temperature is quite low compared to other methods of generating M-N-C electrocatalysts. This could be emphasized as an additional advantage for lower energy intensive synthesis. How do you expect the low temperature to affect the graphitization of the carbon, and will it have an effect on the electrochemistry? Have you systematically varied the synthesis temperature?*

Response: Thank you for the insightful comment. Indeed, the low synthesis

temperature is advantageous for lower energy intensive synthesis and additionally for the achievement of high metal loadings by mitigating the aggregation of metal atoms at high temperature. To study the influence of temperature on the degree of graphitization and electrocatalytic performances, we have varied the synthesis temperature to prepare different samples, which are denoted as Sb-NSCF-600, Sb-NSCF-800 and Sb-NSCF-1000 with the number denoting the synthesis temperature. The structural characterizations of these samples are shown in Figure R1, suggesting that all samples have similar degree of graphitization and are highly defective, as evidenced by the broad XRD peak at $\sim 23.6^\circ$ corresponding to the (002) plane of carbon, the large I_D/I_G ratio in the Raman spectra and the diffused halo ring of the selected-area electron diffraction (SAED). The low crystallinity of these samples could be related to the lack of transition metals (e.g., Fe, Co, Ni, Cu) during high-temperature treatment, which would otherwise serve as catalysts for the formation of graphitic carbon. Without these catalytic metals (as in our case), the graphitization process of carbon typically requires much higher temperature (e.g., $> 2500^\circ\text{C}$) [Carbon 49, 725-729 (2011); Carbon 42, 3049-3055 (2004)]. Therefore, in our synthetic strategy, the graphitization degree is low and it has little dependence on the synthesis temperature (600°C – 1000°C).

The electrocatalytic performances of these samples are displayed in Figure R2. The results show that the catalytic performances are sensitive to the synthesis temperature. Specifically, Sb-NSCF-600, Sb-NSCF-800 and Sb-NSCF-1000 have maximized H_2O_2 selectivity of 97.2%, 89.3% and 93.6%, respectively. Moreover, Sb-NSCF-600 possesses the highest kinetic current density (j_k) of 32.6 mA cm^{-2} at 0.65 V, exceeding those of Sb-NSCF-800 (29.8 mA cm^{-2}) and Sb-NSCF-1000 (4.9 mA cm^{-2}). Furthermore, the analysis of Tafel plots reveals that Sb-NSCF-600 presents a Tafel slope of 29.4 mV dec^{-1} , smaller than those of Sb-NSCF-800 (49.7 mV dec^{-1}) and Sb-NSCF-1000 (42.4 mV dec^{-1}). Therefore, Sb-NSCF-600 was identified as the optimal sample.

Figure R1 and R2 were included in the revised SI as Figure S1 and S2, respectively, and some relevant discussion was added in the revised main text as follows (Page 5 – 6, Line 115 – 121): “After that, Sb_2S_3 -NSC was further calcinated at 600°C under vacuum to remove the Sb_2S_3 nanorods, resulting in the formation of Sb-NSCF. It is noted that the calcination temperature had been optimized by evaluating the catalytic performances of samples calcinated at different temperatures (Fig. S1 and S2). The relatively low calcination temperature of 600°C is beneficial for the lower energy intensive synthesis and the achievement of high metal loadings by mitigating the high-temperature aggregation of metal atoms.”

Figure R1. **a**, XRD patterns of Sb-NSCF-600, Sb-NSCF-800 and Sb-NSCF-1000. **b**, Raman spectra of Sb-NSCF-600, Sb-NSCF-800 and Sb-NSCF-1000. **c – e**, TEM images of Sb-NSCF-600, Sb-NSCF-800 and Sb-NSCF-1000. The inset figures in **c – e** show the SAED of the corresponding sample.

Figure R2. **a**, Electrochemical oxygen reduction polarization curves (solid lines) at a rotation of 1600 rpm and simultaneous H₂O₂ detection currents on the ring electrode (dashed lines) for Sb-NSCF-600, Sb-NSCF-800 and Sb-NSCF-1000 in O₂-saturated 0.1 M KOH electrolyte. **b**, Calculated H₂O₂ selectivity (%) on Sb-NSCF-600, Sb-NSCF-800 and Sb-NSCF-1000 based on the RRDE measurements. **c**, Comparison of H₂O₂ kinetic current density (j_k) at 0.65 V for Sb-NSCF-600, Sb-NSCF-800 and Sb-NSCF-1000. **d**, Comparison of Tafel slopes for Sb-NSCF-600, Sb-NSCF-800 and Sb-NSCF-1000.

Specific comment #5: What is the yield of the synthesis? What batch size can be produced?

Response: Thank you for the question. In a typical synthesis, the starting materials consist of antimony acetate (4.0 g), thioacetamide (4.0 g) and ethylene glycol (4 mL, 4.4 g). The total mass of the starting materials is 12.4 g. At the end of reaction, the collected product weighs ~ 0.0347 g. Accordingly, the yield of the synthesis is ~ 0.28%. The weight loss can be mainly ascribed to the evaporation of the molecular precursors with low boiling point during the calcination process. We have also increased the amounts of starting materials (30.0 g of antimony acetate, 30.0 g of thioacetamide and 33.0 g of ethylene glycol), which resulted in ~ 0.288 g product with a yield of ~ 0.31%. The batch size could be easily enlarged to gram scale by further increasing the feeding amounts of the starting materials.

We have modified the Methods section to include more details regarding the synthesis (Page 16, Line 358 – 369): “Firstly, 4.0 g of antimony acetate and 4.0 g of thioacetamide were dissolved in 4 mL (4.4 g) of ethylene glycol and the mixture was stirred for 24 h to form a reddish homogeneous suspension..... The yield of Sb-NSCF was ~ 0.0347 g. The weight loss can be mainly ascribed to the evaporation of the molecular precursors with low boiling point during the calcination process.”

Specific comment #6: It may be useful to the reader if you include the BET surface area in figure 2i, or in the caption.

Response: As suggested by the reviewer, the BET surface area has been added in the caption of Figure 2i.

Specific comment #7: Page 7, line 137: I don't think the "hollow" nature of the structures is visible in Figure 2b. Furthermore, by definition, I don't think that nanobelts can be hollow, since belts are 2D structures. You should reconsider how to describe these structures.

Response: Thank you for point this out. After reconsideration, we think these structures could be better described as hollow nanofibers. We have modified our discussion about Figure 2b and 2c as follows (Page 7, Line 144 – 148): “Side-view SEM image displays that the flower-like structure is comprised of a cluster of nanofibers (10 – 20 μm in length) bunched at one end and splayed out at the other (Fig. 2b and Fig. S5). The nanofibers are hollow and thin in wall thickness and due to the high aspect ratio and flexibility they tend to collapse together, as revealed by the enlarged SEM image (Fig. 2c)”. Accordingly, the denotations of “Sb-NSCB” and “Sb-NCB” have been changed to “Sb-NSCF” and “Sb-NCF” in the revised manuscript.

Specific comment #8: I don't think "scattered" is the correct way to describe this structure. I would call this a cluster of nanofibers bunched at one end and splayed out at the other. Are these bunches of nanofibers consistent throughout the sample? Are these images representative? Adding a low magnification image would help confirm this (i.e. much lower magnification than currently available in SI).

Response: Thank you for the comment. We have modified the description of the

structure, as discussed in our response to specific Comment #7. The morphology of the nanofibers is consistent throughout the sample and the images provided are representative. As suggested, we have included images at much lower magnifications in the revised SI as Figure S5. These images were also provided below as Figure R3 for your reference.

Figure R3. SEM images of Sb-NSCF at different magnifications.

Specific comment #9: Page 7, line 142: "no presence of metal particles" - You should be more specific - I think you are referring to crystalline metallic nanoparticles.

Response: Thank you for the comment. We have modified the sentence as follows (Page 7, Line 148 – 150): “The TEM image in Fig. 2d confirms this observation and further demonstrates that the surface of Sb-NSCF is clean with no presence of crystalline metal nanoparticles, consistent with the XRD and Raman results.”

Specific comment #10: Why do you think that the pore size is narrowly clustered at 4.2 nm? What is the mechanism of formation of these mesopores? How do they contribute to the electrochemistry?

Response: Thank you for the insightful comment. Based on the pore size distribution analysis (inset in Fig. 2i), the pore size is centered at ~ 3.7 nm rather than 4.2 nm. 4.2 nm was the averaged pore size and it was mistakenly used in the original manuscript. We have modified the sentence as follows (Page 8, Line 168 – 169): “the pore size distributions suggest the formation of mesopores with the pore size centered at ~ 3.7 nm (Fig. 2i).”

The mesopores could be formed via two mechanisms. The first is the removal of the Sb_2S_3 template via evaporation at high temperature. The creation of porous structures in nanocarbons via the evaporation of metal templates during thermal pyrolysis was well demonstrated in previous studies [J. Am. Chem. Soc. 142, 5477-5481 (2020); ACS Catal. 10, 10523-10534 (2020)]. Secondly, the NH_3 gas released from the decomposition of the thioacetamide (one of the starting materials) could etch the amorphous carbon at high temperature, leading to the formation of mesopores [Appl. Surf. Sci. 505, 144574 (2020)].

The mesopores are beneficial for the improvement of mass transport and the exposure of active sites [J. Am. Chem. Soc. 136, 10053-10061 (2014); Adv. Energy Mater. 6, 1502389 (2016)]. We have modified the relevant discussion as follows (Page

8, Line 166 – 173): “The nitrogen adsorption-desorption isotherms determine that Sb-NSCF has a Brunauer-Emmett-Teller (BET) specific surface area of $445.4 \text{ m}^2 \text{ g}^{-1}$ and the pore size distributions suggest the formation of mesopores with the pore size centered at $\sim 3.7 \text{ nm}$ (Fig. 2i). The abundance of mesopores could contribute to the enhanced exposure of active sites and mass transport during catalysis. In addition, Sb-NSCF possesses a large electrochemically active surface area (ESCA) of $59.6 \text{ m}^2 \text{ g}^{-1}$, estimated from the double-layer capacitance (C_{dl}) in the non-Faradaic potential region (Fig. S8).”

Specific comment #11: *Figure 3b: Pyrrolic nitrogen is less thermodynamically stable than graphitic or pyridinic nitrogen, and therefore it is unlikely that this is the most abundant species present in your sample. In addition, if you have 10% Sn in the sample, I'd expect a Sn-N peak to be present too - where is it? It may appear if you reanalyse the deconvolutions and shift the main peaks to appropriate binding energies. Similarly, if you have 24% sulfur in the carbon, a S-N peak should also be assigned in the N1s. Broadly speaking, the assignments could be more likely pyridinic (398.5 eV), Sn-N (~400 eV), graphitic N/hydrogenated pyridinic (~401 eV). I recommend that you check the following paper regarding interpretation of XPS spectra: J. Vac. Sci. Technol. A 38(3) May/June 2020; doi: 10.1116/1.5135923 (especially Fig. 5).*

Response: Thank you for the valuable comment and suggestion. We agree with the reviewer that pyrrolic nitrogen is less thermodynamically stable than graphitic or pyridinic nitrogen. However, we note that the content of pyrrolic nitrogen could be influenced by the structure of the carbon substrate and the synthesis temperature. In our case, the carbon substrate is highly defective and the pyrolysis temperature is relatively low ($600 \text{ }^\circ\text{C}$), which are both beneficial for the formation of pyrrolic nitrogen. This is consistent with previously reports [Nat. Commun. 11, 3884 (2020); Angew. Chem. Int. Ed. 52, 11755-11759 (2013)]. As suggested by the reviewer, we have re-analyzed the XPS N1s spectrum, which was deconvoluted into three peaks (Figure R4), including pyridinic N (398.4 eV), Sb-N/pyrrolic N (400.4 eV) and graphitic N (401.2 eV). The deconvolution of the Sb-N and pyrrolic N species by XPS was difficult due to their similar binding energy [Sci. Adv. 1, e1500462 (2015); J. Vac. Sci. Technol. A 38, 031002 (2020)]. As for the S-N peak, it is difficult to be definitely assigned in the N 1s spectrum and therefore is not included. We have replaced Figure 3b with the updated figure and modified the relevant discussion as follows (Page 9, Line 186 – 188): “The N 1s XPS spectrum in Fig. 3b can be deconvoluted into pyridinic N (398.4 eV), Sb-N/pyrrolic N (400.4 eV) and graphitic N (401.2 eV).”

The following literatures have been added in the Reference section as ref [49] and ref [50] in the revised main manuscript to assist the discussion: Sci. Adv. 1, e1500462 (2015); J. Vac. Sci. Technol. A 38, 031002 (2020).

Figure R4. High-resolution XPS N 1s spectrum.

Specific comment #12: *The electrochemical characterization is essentially meaningless if you don't provide a suitable independent reference sample for comparison. Usually Pt/C is used as a 4 electron reference. As you measured in alkaline electrolyte and are interested in 2 electron ORR, nickel might be more suitable. Even better would be a commercial M-N-C catalyst such as Pajarito Powder.*

Response: Thank you for the valuable suggestion. For the M-N-C catalyst from Pajarito Powder, it is not commercially available in our region. In addition, we have carefully surveyed the literature and found that no common reference samples for the 2-electron ORR were adopted by the researchers in the field. Therefore, we decided to provide the following two reference samples: one is commercial Pt/C and the other is nickel nanoparticles supported on carbon black prepared (denoted as Ni-C) according to the reference of ACS Appl. Mater. Interfaces 12, 17519-17527 (2020). The electrocatalytic performances including the two reference samples are displayed in Figure R5. The results show that Ni-C preferred the 2-electron ORR, while Pt/C favored the 4-electron ORR.

Figure R5 was included in the revised SI as Figure S21 and some relevant discussion was added in the revised main text as follows (Page 11, Line 248 – 251): “As reference samples, the commercial Pt/C catalyst exhibited a diffusion-limited disk current density close to 6 mA cm⁻² with negligible ring current density and H₂O₂ production (Fig. S21), suggesting that it catalyzed the ORR via the 4e⁻ process.”

Figure R5. **a**, Electrochemical oxygen reduction polarization curves (solid lines) at a rotation of 1600 rpm and simultaneous H₂O₂ detection currents on the ring electrode (dashed lines) for Sb-NSCF, Sb-NCF, NSC, Ni-C and Pt/C in O₂-saturated 0.1 M KOH electrolyte. **b**, Calculated H₂O₂ selectivity (%) on Sb-NSCF, Sb-NCF, NSC, Ni-C and Pt/C based on the RRDE measurements.

Specific comment #13: It is not good practice to use a platinum counter electrode in the measurement of nominally platinum-free M-N-C electrocatalysts. In general, a graphite counter electrode should be used to avoid contamination.

Response: Thank you for the comment. As suggested, a graphite counter electrode was used to evaluate the electrochemical performance of the catalyst and the results were compared to those from platinum counter electrode to exclude the possible contamination. As shown in Figure R6, the RRDE polarization curves before and after 3000 cycles obtained with a graphite counter electrode almost overlapped with those with a Pt counter electrode, indicating that the activity, selectivity and stability of the catalyst was not influenced by the Pt counter electrode. We have added some discussion regarding this point in the section of Electrochemical Measurements as follows (Page 17, Line 393 – 395): “To exclude the possible influence of Pt contamination from the Pt counter electrode on the performances, a graphite counter electrode was also used for electrochemical measurements (Fig. S31).”

Figure R6. Electrochemical oxygen reduction polarization curves (solid lines) at a rotation of 1600 rpm and simultaneous H₂O₂ detection currents on the ring electrode (dashed lines) for Sb-NSCF in O₂-saturated 0.1 M KOH electrolyte using Pt and graphite counter electrodes.

Reviewer #2 (Remarks to the Author):

This manuscript developed a main-group metal-nitrogen-carbon catalyst consisting of high-density single Sb atoms supported on N/S-codoped carbon nanobelts (Sb-NSCB) for the electrosynthesis of H₂O₂ via the 2e⁻ ORR. The catalyst was synthesized by a unique Sb₂S₃-templated strategy with high metal-atom loading and hierarchical porous nanostructures and it exhibited exceptional catalytic activity, selectivity and stability. Detailed experimental and theoretical studies were conducted to elucidate the origin for the improved catalytic reactivity. The presented results represent significant achievements in the field of single atom electrocatalysts in terms of new synthetic method, fundamental understanding of the catalytic mechanism and impressive performances and will be attractive to broad audience in materials science and chemical science. Overall, I recommend its publication in Nature Communications after minor revision. Some specific comments are provided as follows:

Response: Thank you for the kind praise about our work and we appreciate the valuable comments that could help to further strengthen the manuscript.

Specific comment #1: *Considering that transition metal-nitrogen-carbon materials such as Co-N-C already exhibited promising catalytic activity for the ORR, what are the potential advantages of developing main-group M-N-C catalysts?*

Response: Thank you for the comment. The potential advantages of developing main-group M-N-C catalysts are listed as follows: (1) The main-group M-N-C catalysts could possess superior durability compared to Co/Fe-N-C due to the Fenton-inactive character of the main-group metals [Nat. Mater. 19, 1215-1223 (2020)]; (2) Different from first-row transition metals, the main-group metals are inactive for catalyzing the graphitization of carbon during the pyrolysis synthesis and thus the carbon substrates in the M-N-C catalysts are typically highly defective and amorphous, which could lead to unique catalytic reactivity via the metal-support interaction; (3) The main-group species possess donor/acceptor frontier orbitals that are separated by modest energy gaps, thus forming frustrated M/C or M/N Lewis acid/base pairs, which were predicted to have high frustrated Lewis pair type reactivity [ACS Appl. Mater. Interfaces 14, 1002-1014 (2022); ChemCatChem 10, 4213-4228 (2018)]. Therefore, main-group M-N-C catalysts have some unique characteristics that make them superior or complementary to transition metal-based counterparts.

We have added some relevant discussion to highlight the potential advantage of developing main-group M-N-C catalysts (Page 7, Line 153 – 156): “The amorphous feature of the carbon substrate in Sb-NSCF can be ascribed to the fact that different from transition metals, the main-group Sb metal is inactive for catalyzing the graphitization of carbon during the pyrolysis synthesis and it could lead to unique catalytic reactivity via the metal-support interaction.”

Specific comment #2: *Increasing the metal loading is important for the applications of SACs. The high-density (10.32 wt%) is a highlight of this work. The authors are suggested to summarize and discuss the synthetic methods of SACs with high metal*

loading in the introduction part: DOI: 10.1007/s12274-020-3244-4; etc.

Response: Thank you for the suggestion. As suggested, we have added some discussion about the synthetic methods of SACs with high metal loading in the introduction part as follows (Page 3 – 4, Line 77 – 81): “To this end, some strategies have been developed to alleviate the aggregation of metal atoms during pyrolysis, such as the metal molecular grafting, spatial confinement, multilayer stabilization and cascade anchoring. However, these strategies often require complex steps and their application to main-group metals are yet to be demonstrated.”

The following literatures have been added in the Reference section as ref [44], ref [45] and ref [46] in the revised main manuscript to assist the discussion: Nano Res. 14, 2418-2423 (2021); Small Struct. 3, 2200041 (2022); Small Methods 4, 1900540 (2020).

Specific comment #3: *The calcination temperature of Sb-NSCB was set at 600 °C under vacuum. Had the calcination temperature been optimized? What is the influence of temperature on the catalytic behaviors?*

Response: Thank you for the question. The calcination temperature had been optimized by evaluating the catalytic performances of samples calcinated at different temperatures (Sb-NSCF-600, Sb-NSCF-800 and Sb-NSCF-1000 with the number denoting the calcination temperature). As shown in Figure R7, Sb-NSCF-600, Sb-NSCF-800 and Sb-NSCF-1000 possess a maximized selectivity of 97.2%, 89.3% and 93.6%, respectively. Also, Sb-NSCF-600 shows the highest kinetic current density (j_k) of 32.6 mA cm⁻² at 0.65 V, compared to those of Sb-NSCF-800 (29.8 mA cm⁻²) and Sb-NSCF-1000 (4.9 mA cm⁻²). Furthermore, the analysis of Tafel plots reveals that Sb-NSCF-600 presents a Tafel slope of 29.4 mV dec⁻¹, smaller than those of Sb-NSCF-800 (49.7 mV dec⁻¹) and Sb-NSCF-1000 (42.4 mV dec⁻¹). These results identified 600 °C as the optimal calcination temperature. Figure R7 was included in the revised SI as Figure S2 and some relevant discussion was added in the revised main text as follows (Page 5 – 6, Line 115 – 119): “After that, Sb₂S₃-NSC was further calcinated at 600 °C under vacuum to remove the Sb₂S₃ nanorods, resulting in the formation of Sb-NSCF. It is noted that the calcination temperature had been optimized by evaluating the catalytic performances of samples calcinated at different temperatures (Fig. S2).”

Figure R7. **a**, Electrochemical oxygen reduction polarization curves (solid lines) at a rotation of 1600 rpm and simultaneous H_2O_2 detection currents on the ring electrode (dashed lines) for Sb-NSCF-600, Sb-NSCF-800 and Sb-NSCF-1000 in O_2 -saturated 0.1 M KOH electrolyte. **b**, Calculated H_2O_2 selectivity (%) on Sb-NSCF-600, Sb-NSCF-800 and Sb-NSCF-1000 based on the RRDE measurements. **c**, Comparison of H_2O_2 kinetic current density (j_k) for Sb-NSCF-600, Sb-NSCF-800 and Sb-NSCF-1000. **d**, Comparison of Tafel slopes for Sb-NSCF-600, Sb-NSCF-800 and Sb-NSCF-1000.

Specific comment #4: The activity of catalyst toward the H_2O_2 reduction reaction is essential in H_2O_2 production. How about the activity of Sb-NSCB toward this reaction?

Response: Thank you for the comment. The activity of the catalyst toward the H_2O_2 reduction reaction was conducted in N_2 -saturated 0.1 M KOH electrolyte containing 3.5 mM H_2O_2 . As shown in Figure R8, compared to Pt/C, Sb-NSCF exhibited an insignificant H_2O_2 reduction current, suggesting its negligible activity toward this reaction. We have modified the relevant discussion as follows (Page 11, Line 251 – 254): “ H_2O_2 reduction reaction measurements in suggested that Sb-NSCF exhibited an insignificant H_2O_2 reduction current compared to Pt/C (Fig. S22), which could avoid the further reduction of the H_2O_2 product.”

The following literature have been added in the Reference section as ref [20] in the revised main manuscript to assist the discussion: Adv. Funct. Mater. 32, 2106886 (2022).

Figure R8. Peroxide reduction reaction for Sb-NSCF and 20 wt% Pt/C in N_2 -saturated 0.1 M KOH electrolyte with or without 3.5 mM H_2O_2 .

Specific comment #5: The BET surface area of Sb-NCB should be provided and compare to that of Sb-NSCB.

Response: Thank you for the suggestion. As suggested, the BET surface area of Sb-NCF was measured and the results were provided in Figure R9. The adsorption-desorption isotherms of Sb-NCF exhibited a typical type-II hysteresis loop at the relative pressure between 0.45 and 0.99, suggesting the presence of abundant mesopores. The BET surface area for Sb-NCF was determined to be $1231.7 \text{ m}^2 \text{ g}^{-1}$, which is larger than that of Sb-NSCF ($445.4 \text{ m}^2 \text{ g}^{-1}$). This could be related to the higher calcination temperature used for preparing Sb-NCF that resulted in more defective sites and porous structures in the carbon substrate. Figure R9 was added as Figure S18 in the revised manuscript.

Figure R9. N_2 adsorption-desorption isotherms and the corresponding pore size distribution of Sb-NCF.

Specific comment #6: The authors are suggested to enhance the discussion on the active sites at atomic scale: DOI: 10.1007/s12274-022-4371-x; etc.

Response: Thank you for the suggestion. We have added more discussion on the active sites at atomic scale in the main text as follows (Page 14 – 14, Line 305 – 309): “Given that the coordinated nitrogen types of the metal centers and the modification of the carbon substrate can both affect the electronic structure and thus catalytic reactivity of

the active metal sites, we considered four structural models of Sb-N-C catalysts adopting the pyridinic and porphyrinic configurations with and without the S dopants.....”. The following literatures have been added in the Reference section as ref [52] and ref [53] in the revised main manuscript to assist the discussion: Adv. Mater. 30, 1800588 (2018); Nano Res. 15, 6888-6923 (2022).

Specific comment #7: Why is the carbon amorphous in the synthesized catalyst?

Response: Thank you for the valuable question. To form crystalline carbon or graphitic carbon by thermal pyrolysis of molecular precursors, transition metals (e.g., Fe, Co, Cu) are typically required to catalyze the graphitization of carbon during the pyrolysis treatment. Without the presence of these transition metals, amorphous carbon is typically formed at pyrolysis temperature lower than 1000 °C [Sci. Adv. 3, e1601821 (2017); Adv. Energy Mater. 8, 1702434 (2018)], and formation of graphitic carbon would require much higher temperature (e.g., > 2500 °C). In our case, the main-group metal of Sb was inactive in catalyzing the graphitization of carbon precursors and additionally the pyrolysis temperature was kept low (600 °C–1000 °C). As a result, the carbon is amorphous in the synthesized catalyst.

Specific comment #8: A more detailed description of the method for determining the generated amounts of H₂O₂ should be needed.

Response: Thank you for the suggestion. As suggested, we have added more details about the method for determining the generated amounts of H₂O₂ as follows (Page 19, Line 439 – 453): “The H₂O₂ concentration was measured by a traditional cerium sulfate Ce(SO₄)₂ titration method based on the mechanism that a yellow solution of Ce⁴⁺ would be reduced by H₂O₂ to colorless Ce³⁺ ($2\text{Ce}^{4+} + \text{H}_2\text{O}_2 \rightarrow 2\text{Ce}^{3+} + 2\text{H}^+ + \text{O}_2$). The concentration of Ce⁴⁺ before and after the reaction can be measured by ultraviolet-visible spectroscopy. Standard Ce(SO₄)₂ solution (0.5 mM) was prepared by dissolving Ce(SO₄)₂ salts into 0.5 M sulfuric acid solution. The calibration curves between absorbance and Ce⁴⁺ concentration were determined by measuring the absorbance at 317 nm of different Ce(SO₄)₂ solutions with known concentrations (0.1 – 0.5 mM) (Fig. S32). After electrolysis for a certain time period, 100 μL of the electrolyte in the cathode chamber after neutralization by 0.5 M sulfuric acid solution was added into the standard Ce(SO₄)₂ titrant solution. Based on the linear relationship between the signal intensity and Ce⁴⁺ concentration, the molar amounts of consumed Ce⁴⁺ after reaction could be obtained. By this approach, the amounts of H₂O₂ produced can be calculated as half the molar amounts of the Ce⁴⁺ consumed”.

REVIEWERS' COMMENTS

Reviewer #2 (Remarks to the Author):

The revised version can be accepted.

I have looked at the authors' response to the points raised by referee #1 and feel they have been adequately addressed.

Reviewer #2 (Remarks to the Author):

General comments: The revised version can be accepted. I have looked at the authors' response to the points raised by referee #1 and feel they have been adequately addressed.

Response: We thank the reviewer for carefully reviewing the revised manuscript and supporting its publication.